# Conformational dynamics and membrane insertion mechanism of B4GALNT1 in ganglioside synthesis

Jack W. J. Welland [1], Henry G. Barrow[1], Phillip J. Stansfeld [2] & Janet E. Deane [1]✉

Glycosphingolipids (GSLs) are crucial membrane components involved in essential cellular pathways. Complex GSLs, known as gangliosides, are synthesised by glycosyltransferase enzymes and imbalances in GSL metabolism cause severe neurological diseases. B4GALNT1 synthesises the precursors to the major brain gangliosides. Loss of B4GALNT1 function causes hereditary spastic paraplegia, while its overexpression is linked to cancers including childhood neuroblastoma. Here, we present crystal structures of the human homodimeric B4GALNT1 enzyme demonstrating dynamic remodelling of the substrate binding site during catalysis. We show that processing of lipid substrates by B4GALNT1 is severely compromised when surface loops flanking the active site are mutated from hydrophobic residues to polar. Molecular dynamics simulations support that these loops can insert into the lipid bilayer explaining how B4GALNT1 accesses and processes lipid substrates. By combining structure prediction and molecular simulations we propose that this mechanism of dynamic membrane insertion is exploited by other, structurally distinct GSL synthesising enzymes.

Sphingolipids are an important class of lipids found in the outer leaflet of the plasma membrane. They possess a ceramide backbone buried in the membrane, which is composed of a sphingosine and fatty acyl chain (Fig. 1A). Glycosphingolipids (GSLs) are formed by modifying the ceramide backbone with polar carbohydrate headgroups that are exposed on the cell surface. GSLs can be further modified to contain a sialic acid group, these are collectively known as gangliosides. GSLs play crucial roles in cell function by forming membrane microdomains (also known as lipid rafts) containing cholesterol and specific membrane proteins[1]. These membrane microdomains form essential signalling platforms that control processes ranging from neuronal development and synapse formation to immune receptor function[2]. When GSL levels are perturbed, this causes a wide range of severe diseases including lysosomal storage diseases, leukodystrophies, hereditary spastic paraplegias and cancer[2–4].

The synthesis of gangliosides occurs in the ER/Golgi in a stepwise manner involving several membrane-bound glycosyltransferase enzymes[5,6]. The localisation, activity and abundance of these glycosyltransferases is tightly regulated in cells to ensure different ganglioside levels are correctly maintained. β−1,4-N-Acetyl-Galactosaminyltransferase 1 (B4GALNT1) is a key enzyme of the ganglioside synthetic pathway, responsible for the synthesis of complex gangliosides[7]. B4GALNT1 catalyses the synthesis of several gangliosides, including GM2 and GD2 from their respective precursors GM3 and GD3 (Fig. 1B). These products are the essential precursors of the major brain gangliosides GM1(a), GD1a, GD1b and GT1b. B4GALNT1 catalyses the transfer of an N-acetylgalactosamine (GalNac) from a Uridine diphosphate-GalNac donor (UDP-GalNac) onto the glycan headgroup of the ganglioside acceptor substrate (Fig. 1C).

Loss of function of B4GALNT1 causes a complex early-onset form of hereditary spastic paraplegia 26 (HSP26)[8,9]. This form of HSP not

[1]Cambridge Institute for Medical Research, Department of Clinical Neuroscience, University of Cambridge, Cambridge, UK. [2]School of Life Sciences and Department of Chemistry, Gibbet Hill Campus, The University of Warwick, Warwick, UK. ✉e-mail: jed55@cam.ac.uk

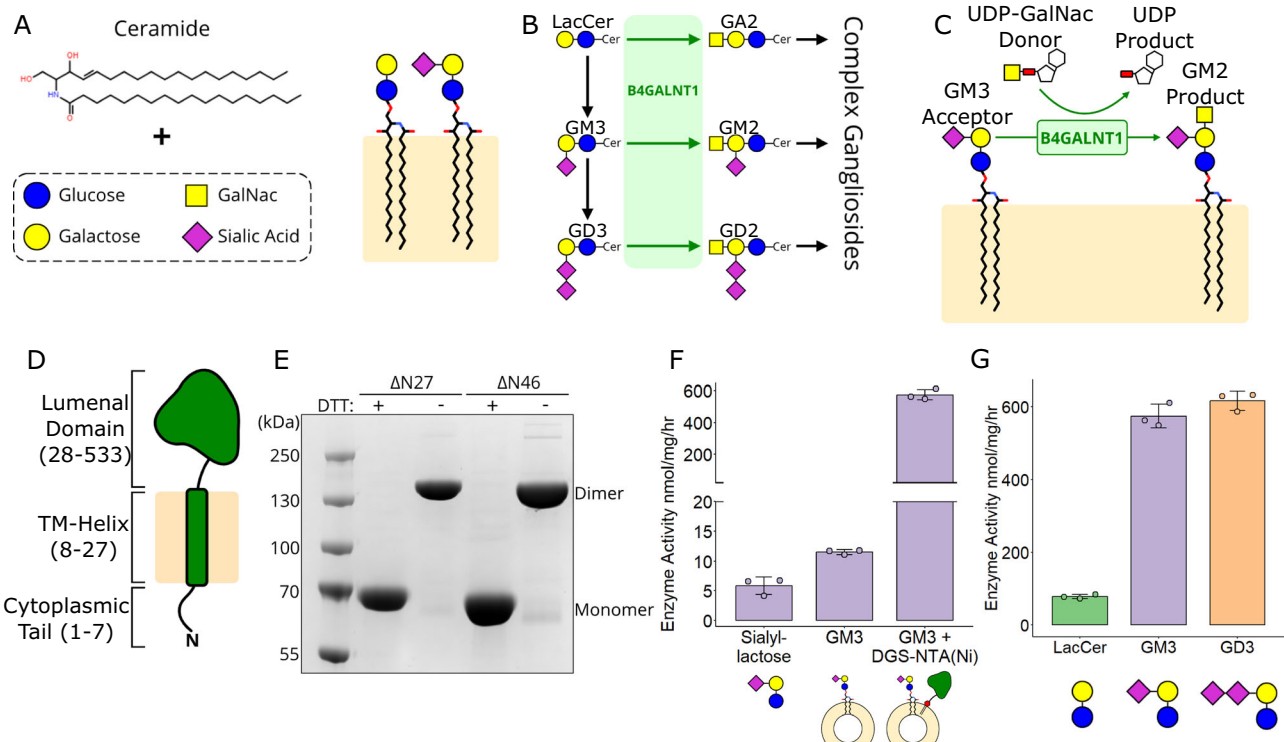

**Fig. 1 | Ganglioside synthesis and B4GALNT1 catalytic activity. A** Schematic diagram illustrating that glycosphingolipids (GSLs) are composed of a ceramide backbone with the addition of glycosylated headgroups. Sialylated GSLs are known as gangliosides. **B** Schematic diagram of part of the ganglioside synthetic pathway illustrating the step B4GALNT1 plays in GSL synthesis. **C** Schematic diagram of the B4GALNT1 donor and acceptor substrate structures. **D** The topology of full-length B4GALNT1 with the numbering for domain boundaries illustrated. **E** SDS-PAGE of B4GALNT1 expression constructs in the presence and absence of reducing agent (DTT), demonstrating B4GALNT1 is a constitutive disulfide-mediated dimer. This gel is representative of N > 10 gels. **F** Enzyme activity of soluble B4GALNT1 lumenal domain against isolated GM3 glycan headgroup (sialyllactose) and lipidated GM3 in liposomes (10% GM3, 90% PC) compared to activity against GM3 lipid when B4GALNT1 is tethered to the liposome membrane via the His$_{10}$ tag to DGS-NTA (Nickel) lipids (10% GM3, 2% DGS-NTA, 88% PC). **G** Relative activity of B4GALNT1 against the different lipid substrates lactosylceramide (LacCer), GM3 and GD3 using the membrane tethered assay, including DGS-NTA lipids in the liposomes (10% Ganglioside, 2% DGS-NTA, 88% PC). For activity assays, the mean is displayed ± standard deviation (SD) for N = 3 biological replicates. SD was calculated using the ggpubr package in R.

only presents with progressive spasticity and weakness in the lower limbs but also cognitive impairment and peripheral neuropathy. Overexpression or increased activity of B4GALNT1 is associated with several cancers including childhood neuroblastoma[10–12]. The B4GALNT1 product GD2 is a key antibody-mediated chemotherapeutic target in neuroblastoma[13–15]. B4GALNT1 is also upregulated in hepatocellular carcinoma (HCC) and plays a role in immunosuppression, with the silencing of B4GALNT1 improving tumour-killing in mouse models treated with programmed cell death protein 1 targeting strategies[16]. Therefore, inhibition of B4GALNT1 may represent an auxiliary therapeutic strategy when coupled to immunotherapy in HCC.

CD38-mediated upregulation of B4GALNT1 has recently been implicated in the pathology of Systemic Lupus Erythematosus (SLE)[17]. The resulting increase in GM2 at the cell membrane was associated with an increased Ca$^{2+}$ response and consequent ER stress, causing decreased cytokine production. The authors suggest that regulating ganglioside expression on T cell membranes could provide a therapeutic strategy for SLE patients. B4GALNT1 is also a potential target for substrate reduction therapy in GM2 gangliosidoses (Tay-Sach's, Sandhoff, AB variant), whereby inhibition of B4GALNT1 would result in the accumulating substrate not being synthesised. This would be a more specific approach compared to other substrate reduction strategies trialled against GM2 gangliosidoses utilising the glucosylceramide synthase inhibitor miglustat[18].

There are no experimental structures available for any GSL synthetic enzymes, limiting our understanding of how lipid substrates are

processed, how substrate ligands bind and how disease-associated mutations may impact glycosyltransferase function. Previous studies highlight ambiguities regarding the oligomeric state of B4GALNT1 and oligomer organisation. Early mass spectrometry-based studies suggest that B4GALNT1 forms an anti-parallel, disulphide-mediated dimer[19], while recent studies predicting the impact of HSP- and cancer-associated missense mutations used a monomeric AlphaFold2 (AF2) model of B4GALNT1[9,20].

Here, we present multiple structures of the B4GALNT1 dimer demonstrating conformational changes upon binding of UDP substrates and product. These structures support a catalytic mechanism that involves dynamic remodelling of the substrate binding site and provide insights into the functional consequences of HSP26-associated mutations. We also demonstrate that B4GALNT1 activity depends on hydrophobic residues present on two surface loops flanking the active site. Molecular dynamics simulations support that these loops insert into the lipid bilayer, explaining how B4GALNT1 accesses and processes membrane-embedded substrates. Despite the structural diversity of GSL synthetic enzymes, we demonstrate that several other enzymes in this pathway exploit similar mechanisms for accessing lipid substrates.

## Results

### B4GALNT1 is a constitutive dimer that efficiently processes membrane-embedded lipids

B4GALNT1 is a type-II membrane protein with a predicted domain structure consisting of a short (7-residue) N-terminal cytoplasmic

sequence followed by a transmembrane helix and a 56 kDa lumenal domain (Fig. 1D). B4GALNT1 is localised to the trans-Golgi network (TGN), allowing the lumenal domain to access lipid substrates on the inner leaflet of the TGN membrane[21,22]. Based on AF2 predictions, two constructs of the human B4GALNT1 lumenal domain were designed: one encompassing the entire lumenal domain (ΔN27), and one truncating residues from the N-terminus that were predicted to be disordered (ΔN46). These proteins were expressed and purified using HEK293-F cells, allowing for the preservation of native folding and N-linked glycosylation, of which B4GALNT1 has 3 predicted sites. These constructs expressed well and purified as constitutive, disulphide-mediated dimers as demonstrated by non-reducing SDS-PAGE (Fig. 1E).

To ensure the purified protein was functional, a series of activity assays were developed. Activity was monitored via a two-step UDP-Glo™ assay where the UDP product from the B4GALNT1 reaction is converted to ATP and used as a substrate for luciferase, allowing detection by luminescence (Supplementary Fig. S1). Initial assays using the GM3 glycan headgroup mimic sialyllactose produced very low activity (Fig. 1F). The assay was modified to incorporate the endogenous, lipidated substrate GM3 within liposomes. Addition of the soluble B4GALNT1 lumenal domain resulted in ~2-fold higher activity than for the substrate headgroup alone, but activity remained relatively low (Fig. 1F). As B4GALNT1 is natively membrane bound via a transmembrane helix, DGS-NTA lipids were added to the liposome mix to bind the N-terminal His tag of the protein to mimic the endogenous membrane association of B4GALNT1. Using these DGS-NTA-containing liposomes, B4GALNT1 activity was ~60-fold higher than when B4GALNT1 is free in solution (Fig. 1F).

The activity profile of B4GALNT1 does not display standard Michaelis-Menten kinetics, potentially due to the substrates being embedded in liposomes, the enzyme being tethered to the membrane and/or related to the dimeric nature of the enzyme (Supplementary Fig. S2A). The complexity of interpreting these data is supported by studies exploring how reduced dimensionality, such as membrane tethering, increases the relative concentration of a protein and drives more frequent interaction with lipid substrates[23]. This means that standard kinetic parameters are very difficult to determine, instead, relative activity can be compared using an endpoint assay with a time point and enzyme concentration where the activity remains linear (Supplementary Fig. S2B). Comparison of B4GALNT1 activity against its three main substrates shows that the lipid headgroups containing sialic acid moieties, GM3 and GD3, are better substrates than the smaller lactosylceramide (LacCer) substrate (Fig. 1G). This suggests that these larger headgroups can bind more tightly within the active site, effectively possessing a lower $K_m$ (Supplementary Fig. 2A).

Previous work using the LacCer, GM3 and GD3 lipid substrates dissolved in detergents reported $V_{max}$ values of 0.7 nmol/mg/hr, 5.8–6.2 nmol/mg/hr and 3.5–3.9 nmol/mg/hr, respectively[7]. The assay developed here, using the same substrates, demonstrated activities of 79, 570 and 620 nmol/mg/hr, respectively, supporting that this alternative liposome-based assay with tethered enzyme may represent a better approach for studying B4GALNT1 activity.

## B4GALNT1 possesses two dynamically-remodelled active sites

The B4GALNT1 lumenal domain construct ΔN46 was crystallised for structure determination. Initial crystals possessed no ligands in the active site and diffracted to relatively low resolution, 3.6 Å. Microseeding during crystallisation combined with inclusion of manganese (Mn) in purification buffers improved the resolution of the structure to 2.94 Å for an apo-complex. Inclusion of UDP ligands during crystal harvesting improved the resolution to 2.75 Å with UDP-GalNac substrate and 2.67 Å with UDP product (Table 1). The structure was solved using molecular replacement with an AF2-generated model of the dimer.

**Table 1 | Data collection and refinement statistics***

|  | Apo B4GALNT1 | B4GALNT1 with UDP-GalNac | B4GALNT1 with UDP |
|---|---|---|---|
| Data collection |  |  |  |
| Beamline | Diamond I24 | Diamond I24 | Diamond I24 |
| Wavelength (Å) | 0.6199 | 0.6199 | 0.6199 |
| Space group | $P2_1$ | $P2_1$ | $P2_1$ |
| Cell dimensions |  |  |  |
| a,b,c (Å) | 60.46, 136.49, 87.43 | 60.10, 134.92, 86.58 | 60.18, 133.53, 86.92 |
| β (°) | 96.08 | 95.91 | 94.83 |
| Resolution (Å) | 136.49–2.94 (3.10–2.94) | 54.66–2.75 (2.88–2.75) | 66.76–2.67 (2.79–2.67) |
| *Rmerge* | 0.112 (1.61) | 0.186 (1.442) | 0.127 (1.107) |
| *Rpim* | 0.069 (0.983) | 0.114 (0.964) | 0.079 (0.784) |
| CC1/2 | 0.999 (0.521) | 0.992 (0.389) | 0.989 (0.477) |
| *I/σI* | 9.9 (1.1) | 6.2 (1.0) | 8.0 (1.1) |
| Completeness (%) | 100.0 (100.0) | 99.5 (96.3) | 99.3 (94.2) |
| Multiplicity | 7.1 (7.2) | 7.0 (6.3) | 6.8 (5.6) |
| Refinement |  |  |  |
| Resolution (Å) | 86.936–2.960 | 54.66–2.75 | 66.76–2.674 |
| No. reflections | 29240 | 35558 | 38317 |
| *Rwork/Rfree* | 0.2019/0.2540 | 0.1887/0.2248 | 0.1907/0.2326 |
| *No. atoms* |  |  |  |
| Protein (+Ligands) | 7053 | 7516 | 7322 |
| Water | 14 | 74 | 27 |
| *B-factors* |  |  |  |
| Protein (+Ligands) | 119.01 | 75.80 | 80.05 |
| Water | 78.91 | 57.10 | 56.70 |
| Ramachandran |  |  |  |
| Favoured (%) | 96.7 | 97.4 | 96.7 |
| Outliers (%) | 0.0 | 0.0 | 0.0 |
| *r.m.s. deviations* |  |  |  |
| Bond lengths (Å) | 0.0068 | 0.0078 | 0.0086 |
| Bond angles (°) | 1.8700 | 1.8840 | 2.0060 |
| PDB entry | 9H6J | 9H6K | 9H6L |

*Data in parentheses relate to the highest resolution shell.

Each monomer of the B4GALNT1 homodimer possesses two domains: an N-terminal region encompassing residues 130 to 257 that forms the non-catalytic domain (the 'lower' domain in Fig. 2A) while the C-terminal region, residues 258–533 forms much of the predicted catalytic domain (the 'upper' domain in Fig. 2A). The N-terminal residues 51–129 wrap across the dimer and contribute loops and helices to both the upper and lower domains (Fig. 2B and Supplementary Fig. S3). During refinement, it became clear that the AF2 dimer differed from the experimental structure. At residues 141–144, the AF2 mainchain from one monomer crosses over to the second monomer, while the electron density in this region clearly demonstrates this was not correct (Supplementary Fig. S4). This rethreading has no impact upon the overall shape of the molecule but does change which chains form disulphide bonds. B4GALNT1 possesses three potential disulphide bonds, C80-C412, C82-C529 and C429-C476. In the experimental structure, all are intramolecular disulphide bonds, while in the AF2 model, due to the rethreading, C80-C412 and C82-C529 are intermolecular bonds. Intermolecular disulphide bond formation would be consistent with the non-reducing SDS-PAGE data showing that B4GALNT1 forms a disulphide-mediated dimer. However, the electron

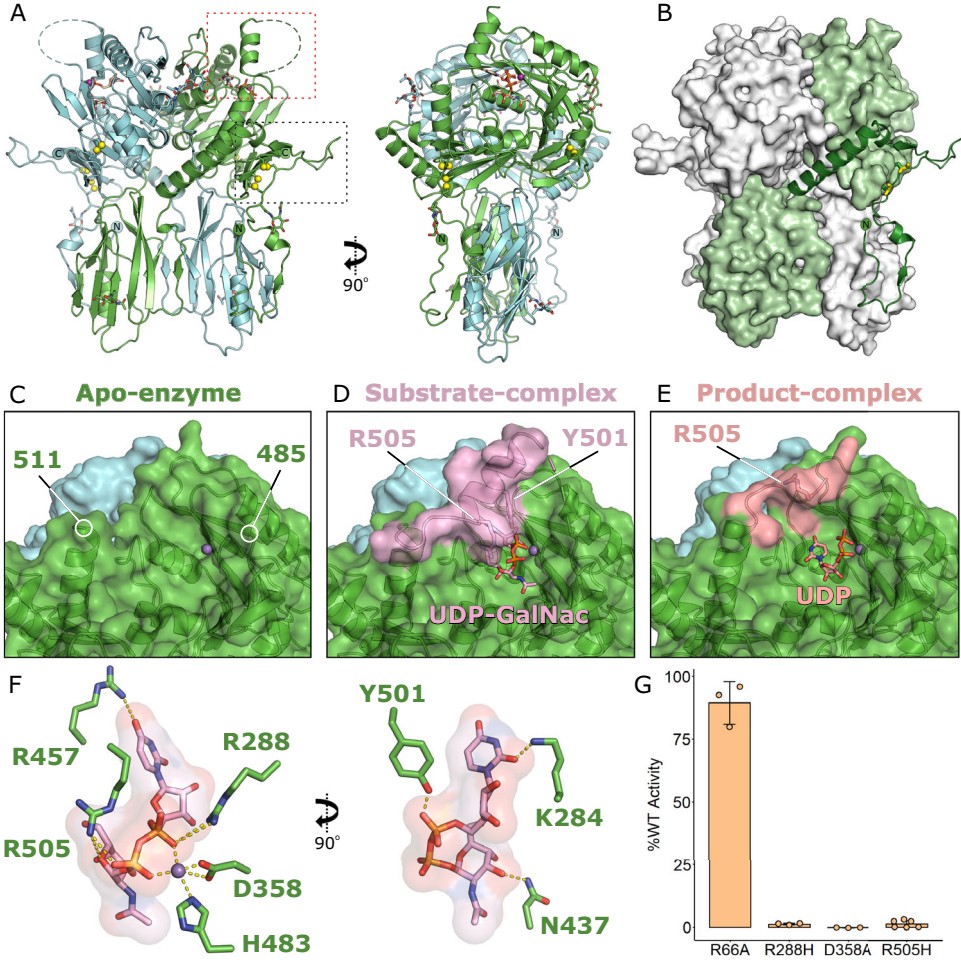

**Fig. 2 | B4GALNT1 structures reveal the localisation and dynamic rearrangements of the substrate donor ligand binding site. A** Ribbon diagram of the X-ray crystal structure of the B4GALNT1 dimer showing two orientations rotated by 90°. Two surface loops are highlighted (dotted boxes), one disordered loop (dashed line) and one well-ordered loop. Disulphide bonds are illustrated (yellow spheres), as are glycans and UDP-GalNac ligand (sticks). **B** The two chains of the dimer, illustrated in green and white, are intertwined. The N-terminal region of one chain (green ribbon) wraps across the surface, is stabilised by disulphide bonds (yellow spheres) and leads into the 'lower' non-catalytic domain that then extends across to the opposite side of the dimer to form the 'upper' catalytic domain. **C** Surface representation of the B4GALNT1 active site illustrating that in the Mn-bound (purple sphere) structure, without UDP ligands, the pocket is open and residues

486–510 are disordered. **D** Upon binding UDP-GalNac, the disordered loop containing residues Y501 and R505 forms a helix (pink ribbon and surface), defining the shape of the ligand binding pocket. **E** Binding of the UDP product in the active site releases the helix seen in the UDP-GalNac structure, with this loop adopting a different conformation (orange ribbon and surface), opening up the binding pocket again. **F** Several residues contribute to the binding of UDP-GalNac and Mn ligands in the active site. For clarity, two orientations of the UDP-GalNac are shown rotated by 90°. Residues that directly bind the ligand or Mn ion are shown (green sticks). **G** Activity assays with B4GALNT1 variants including mutations of key active site residues R288, D358 and R505 (10% GD3, 2% DGS-NTA, 88% PC liposomes). The mean is displayed ± SD for $N = 3$ biological replicates.

---

density does not support this conformation. In both cases, these two disulphide bonds contribute to the orientation and/or stabilisation of a long N-terminal loop that protrudes from the surface of the enzyme, encompassing residues 83–98 (Fig. 2A, black box). In the crystal structure, this loop inserts into an adjacent active site, suggesting that the orientation of this loop captured here may be artificially rigid and in solution, it may possess greater flexibility.

In the apo structure of B4GALNT1, a long loop encompassing residues 486–510 was completely disordered (Fig. 2C). Upon binding of UDP-GalNac, residues 495–510 from this loop become ordered, forming a helix that lies over the active site. Within this helix, residues R505 and Y501 directly bind the phosphate groups of the UDP-GalNac substrate (Fig. 2D). This observation suggests that this loop dynamically rearranges to allow binding and release of UDP-GalNac during catalytic cycling. In further support of this flexibility, the presence of UDP product results in partial ordering of this loop adopting conformations between the apo and substrate complex structures (Fig. 2E). Although there is clear electron density in this region, it is

difficult to model a single conformation for this loop in either monomer in this product complex supporting its dynamic movement even *in crystallo*. Also of note in the product structure is that the UDP moiety has moved within the active site, with the phosphate groups flipping orientation and the uracil group moving to where the GalNac had bound (Supplementary Fig. S5). It remains unclear whether this orientation is a crystal artefact or represents the trajectory of UDP release from the enzyme active site.

B4GALNT1 is the only validated human protein in CAZY glycosyltransferase family 12 and is an inverting GT-A type glycosyltransferase[24,25]. The structure of B4GALNT1 with the UDP-GalNac substrate confirms the location of the active site and the identity of key substrate binding residues (Fig. 2F). Glycosyltransferase enzymes often possess a conserved DxD motif where the aspartic acids are required for binding UDP and the Mn ion[24]. In B4GALNT1, these are D356 and D358. In addition to these residues and Y501 and R505 mentioned above, the sidechains of K284, R288 and R457 all directly bind the UDP-GalNac substrate. Structural homology searches using

the full-length B4GALNT1 did not identify any structures with homology to the overall fold but were able to identify structures with similarity to the core of the catalytic domain. Despite this core structural homology, analysis using consurf[26] shows that the sequence conservation is primarily restricted to the UDP-sugar binding site and adjacent pocket regions, with the more distal parts of the pocket poorly conserved (Supplementary Fig. S6). Several features of the fold, including the R505 helix and additional surface loops, were not present in other glycosyltransferase structures. A structure-based sequence alignment of AlphaFold models of the other CAZY GT12 family members and the top structural homologues identified by DALI highlights some conservation of key active site residues (Supplementary Fig. S7). Specifically, 5 residues involved in UDP-GalNac binding (K284, N437, R457, Y501 and R505) are only conserved in the GT12 family, while 3 other regions (R288, D358 region and H483) are partially conserved in other structurally related glycosyltransferases. Given the diversity of complex glycan substrates, it is not unexpected to see substantial diversity in the active site residues of glycosyltransferases.

### The impact of disease-causing mutations on B4GALNT1 activity

Several missense mutations of B4GALNT1 have been linked to the development of HSP26 (Table 2). Based on the structures of B4GALNT1, three of these variants (K284N, R505H and R505C) will be catalytically inactive as they directly bind the UDP-GalNac substrate (Fig. 2F). Three others (D433 A, P453R and R472P) will destabilise dimer formation as they either lie in the dimer interface or directly stabilise residues at the interface (Fig. 3A). Three more variants (R300C, S475F and R519P) are highly likely to result in misfolding of B4GALNT1 as they are buried within the fold, engage in hydrogen bonding networks and each of the mutations abolishes the ability to form these bonds (Fig. 3B). Furthermore, in the case of S475F the large hydrophobic sidechain would not pack correctly in the available space and for R519P the proline will break the helical fold in this region of the structure. Analysis of the NCBI ClinVar[27] and gnomAD[28] databases, identifies over 100 different missense variants that have been identified in patients with spastic paraplegia but are currently of uncertain significance (Supplementary Table 1). Three of these variants of uncertain significance (K350E, R260G and E201Q) form salt bridges at the dimer interface, suggesting they may contribute to disease by disrupting dimer formation. Also of note in these databases was R288H that is a residue directly involved in binding UDP-GalNac substrate, strongly suggesting that this will be a pathogenic variant.

To test the impact of UDP-GalNac-binding variants, R288H and R505H, as well as a catalytically dead variant mutating one of the key aspartate residues required for binding Mn, D358A, were produced. An additional mutation, R66A, was produced that was predicted not to interfere with folding, dimerisation or activity. All four variants were successfully expressed and purified, eluted similarly following size-exclusion chromatography and possessed equivalent melting temperatures by thermal shift assay, demonstrating that they were all correctly folded (Supplementary Fig. S8A, B). The two variants predicted to be disease-causing, R288H and R505H, possessed < 2% of the activity of the wild-type enzyme, equivalent to that of the catalytically dead variant D358A (Fig. 2G). These data confirm the active site structure and identify that these disease variants cause pathology via loss of enzymatic activity caused by inability to bind substrate.

### Mutation of hydrophobic surface loops impacts B4GALNT1 activity

The structural and functional data shown here explain how the donor ligand, UDP-GalNac, binds the active site and that this binding partially orders a loop of the enzyme that lies over the active site. To understand how the acceptor lipid substrates may bind, the B4GALNT1 structure was compared to protein glycosyltransferases with peptide ligands bound. This suggested the lipids are likely to bind in the pocket near the UDP-GalNac bounded by two surface loops (Fig. 4A). As described above, one of these loops (residues 83–98) extends from the N-terminal domain and is partially oriented via disulphide bonds while the other loop (residues 486–494) remains disordered in all crystal structures. Both loops contain strings of hydrophobic residues (LPLPF and LPW, respectively), suggesting they might help orient lipid substrates in the active site by inserting into membranes and/or interacting with hydrophobic acyl chains (Fig. 4A, zoom panel).

To test the functional importance of these loops for processing lipid substrates, hydrophobic residues in these loops, F93 and W491, were mutated to polar side chains to interfere with potential membrane insertion. The variants F93R and W491N were expressed, purified and shown to be correctly folded (Supplementary Fig. S8C, D). Activity assays using lipid substrates in liposomes and tethered B4GALNT1 clearly demonstrate that these mutations have a significant impact on catalytic activity, reducing it to < 25% of WT (Fig. 4B). Significant reduction in activity is also seen when B4GALNT1 is not tethered to the liposome membrane (Fig. 4C). In contrast, these mutations did not have a significant effect on activity against the non-lipidated substrate sialyllactose (Fig. 4C), supporting that these loops are crucial for the processing of lipid substrates in membranes.

To further test the importance of these loops, 'rescue' mutations, F93W and W491F, were introduced that maintain the hydrophobic nature of these residues. W491F effectively returned enzyme activity for lipidated substrates to almost 100% (Fig. 4D). Interestingly, mutation of F93 to a tryptophan substantially increased activity, making B4GALNT1 2–4 times more active against lipidated substrates (Fig. 4D). These activity data combined with the structural data demonstrate that although these residues are not directly part of the catalytic mechanism, their hydrophobic properties are required for efficient substrate processing.

### B4GALNT1 surface loops interact with and insert into membranes

To test if the B4GALNT1 lumenal domain interacts with membranes and if so, do the hydrophobic surface loops contribute to this binding, molecular dynamics (MD) simulations were carried out. Coarse-grain MD simulations using the B4GALNT1 lumenal domain structure and a membrane composed of phosphatidylcholine (PC) and 2 molecules of GM3 in either leaflet (equivalent of 1%) demonstrates that during a 5 μs simulation, the B4GALNT1 lumenal domain approaches the membrane and consistently makes sustained contacts with the PC lipids (Fig. 5A). When associated with the membrane, the two surface loops containing F93 and W491 insert into the lipid bilayer (Fig. 5B). A per residue analysis of the B4GALNT1 membrane interactions across 25 simulations highlights that the loops containing F93 and W491 form most of

### Table 2 | B4GALNT1 missense mutations linked to HSP26

| Mutation | Predicted Impact Based on Structure | References |
|---|---|---|
| K284N | Catalytic: directly binds UDP-GalNac substrate | 9,58–60 |
| R300C | Misfolding: stabilises the fold in the non-catalytic domain | 9,58,60,61 |
| D433A | Destabilises dimer: bonds to H420 that pi-stacks with Y230 at the dimer interface | 9,58–62 |
| P453R | Destabilises dimer: mediates a backbone turn at the dimer interface | 9 |
| R472P | Destabilises dimer: forms H-bonds at the dimer interface | 62 |
| S475F | Misfolding: buried, pocket can't fit bigger, hydrophobic F sidechain | 63 |
| R505H | Catalytic: directly binds UDP-GalNac substrate | 9,58–60 |
| R505C | Catalytic: directly binds UDP-GalNac substrate | 62 |
| R519P | Misfolding: Pro breaks helix and breaks H-bond | 9 |

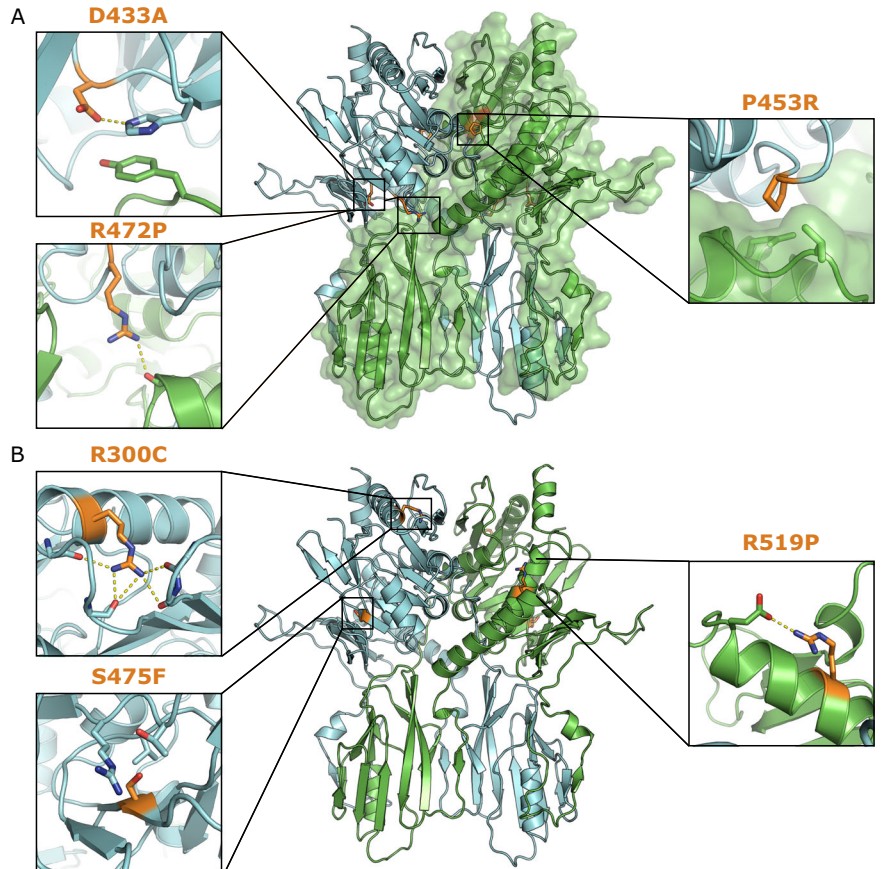

**Fig. 3 | Structure-based prediction of the impact of B4GALNT1 mutations.**
**A** Illustration of the location and interactions of three HSP-causing mutations that lie at the B4GALNT1 dimer interface. D433A stabilises a Histidine that pi-stacks with a tyrosine residue in the adjacent monomer. Loss of this sidechain would desta-bilise these interactions. The sidechain of R472 forms a hydrogen bond with a backbone carboxyl in the adjacent monomer. Loss of this interaction, as well as the potential backbone rearrangement caused by mutation to proline, would destabi-lise this interface. P453 lies within a hydrophobic pocket at the dimer interface, mutation to arginine would severely disrupt this interaction. **B** Illustration of the location and interactions of three HSP-causing mutations that are predicted to cause misfolding or destabilise the structure. The sidechain of R300 engages in several hydrogen bonds that stabilise loops adjacent to the R300-containing helix. S475 lies within the fold of B4GALNT1 without space to accommodate the larger phenylalanine sidechain. The R519 sidechain engages in hydrogen bonding, and loss of this, combined with the mutation to proline, will destabilise the helix and fold in this region.

the membrane contacts (Fig. 5C, D) while another small region near the active site at residue S390 also contacts the membrane. In this membrane-associated orientation, one of the active sites within the dimer is positioned over the surface of the membrane, compatible with binding and processing of the glycosylated headgroup of a lipid substrate. The other active site in the dimer faces away from the membrane.

MD simulations were repeated using modified B4GALNT1, intro-ducing the point mutations F93R and W491N, that significantly impact enzyme activity against lipid substrates. Over the course of these simulations, the B4GALNT1 lumenal domain still associated with membranes although with different apparent kinetics interpreted through protein-PC contacts (Fig. 5E). The per residue analysis of these trajectories also shows that the contact with the membrane was changed (Fig. 5F). Compared to WT B4GALNT1, the F93R and W491N variants make fewer contacts with membrane lipids, with these changes being driven by reduced interaction of the mutated loops them-selves. Over the trajectory of the simulations, the F93R is more dramatically reduced in its overall lipid contacts compared to the W491N, suggesting it cannot associate as effectively with the mem-brane (Fig. 5E).

In these MD simulations, B4GALNT1 interacts with the PC lipids in the membrane, consistent with the binding being driven by hydro-phobic interactions between the two associating loops and the

membrane. The model membrane also contains molecules of GM3, and during the simulations, this is captured in the active site. Analysis of all frames from 25 simulations using PyLipID[29] identified repre-sentative bound poses for GM3 in the highest occupancy binding site across the simulations. The top-ranked GM3-binding pose was con-verted to atomistic with CG2AT[30] and used as the starting state for all-atom simulations. During the 500 ns simulation, the GM3 molecule samples poses that illustrate how it can bind deep in the active site (Supplementary Movie M1). This binding also demonstrates how important the substantial burial of the B4GALNT1 hydrophobic loops into the membrane is to facilitate complete access to the active site for the GM3 lipid (Fig. 5G).

To compare how membrane tethering influences the orientation of the catalytic domain or insertion of the hydrophobic loops, a model was made of the full-length protein, including the transmembrane helix. Using this model, the same loops insert into the membrane with the lumenal domain adopting a similar orientation to that when it is untethered (Supplementary Fig. S9). The main difference in the full-length MD simulations is how rapidly the loops insert into the mem-brane and the relative contribution each of these regions makes during membrane association.

The orientation of the active site relative to the membrane and the insertion of the flanking, hydrophobic loops explain how the B4GALNT1 enzyme can cleave lipid substrates while they remain

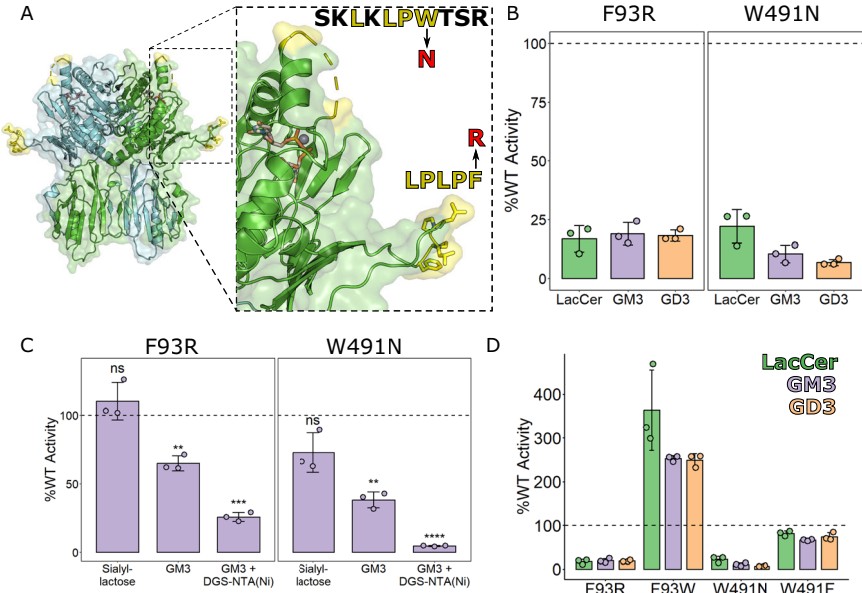

**Fig. 4 | Hydrophobic surface loops near the ligand binding site contribute to B4GALNT1 activity. A** Ribbon diagram of the B4GALNT1 structure with hydrophobic loop regions flanking the active site highlighted (yellow). Zoom panel: The sequences of the loops are shown with hydrophobic residues highlighted (yellow) and the key polar mutations indicated (red). **B** Relative enzyme activity of polar loop mutants F93R and W491N against three different lipid substrates (10% Ganglioside, 2% DGS-NTA, 88% PC liposomes). **C** Relative enzyme activity of F93R and W491N polar loop mutants using the different GM3-based activity assays testing the headgroup alone (sialyllactose) and the membrane-embedded GM3 lipid without tethered B4GALNT1 (10% GM3, 90% PC liposomes) and with the DGS-NTA lipid for tethering of B4GALNT1 (10% GM3, 2% DGS-NTA, 88% PC liposomes). Significance

was calculated via a two-tailed Student's $t$ test implemented in R with the rstatix package. ** $p < 0.01$ *** $p < 0.001$ **** $p < 0.0001$. F93R vs WT Sialyllactose: $p = 0.319$, F93R vs WT GM3: $p = 0.008$, F93R vs WT GM3 + DGS-NTA(Ni): $p = 6.5e\text{-}4$. W491N vs WT Sialyllactose: $p = 0.166$, W491N vs WT GM3: $p = 0.006$, W491N vs WT GM3 + DGS-NTA(Ni): $p = 9.12e\text{-}6$. **D** Relative enzyme activity of polar and 'rescue' mutants at F93 and W491 against three lipid substrates using the B4GALNT1-tethered assay with DGS-NTA-containing liposomes (10% Ganglioside, 2% DGS-NTA, 88% PC liposomes). For activity assays, the mean is displayed ± SD for $N = 3$ biological replicates. In panels (**B**) and (**D**), the same data for F93R and W491N has been re-plotted for clarity with and without the rescue mutants.

embedded in a membrane rather than requiring a lipid-binding transfer protein, such as is needed for several GSL hydrolytic enzymes[31,32].

## Hydrophobic loop insertion as a mechanism to synthesise other GSLs

The structure of B4GALNT1 represents a novel experimental structure of a GSL synthetic enzyme. This structure enabled the prediction of how substrates bind and how surface loops can influence enzyme activity against lipid substrates by inserting into membranes. Inspection of AF2 models of other GSL synthetic enzymes revealed potential functional similarities regarding hydrophobic surface loops flanking the predicted active site. One particularly compelling candidate was ST3GAL5, responsible for synthesising the B4GALNT1 substrate GM3 from LacCer (Fig. 6A). Although ST3GAL5 and B4GALNT1 do not possess structural similarity, ST3GAL5 also possesses hydrophobic surface patches surrounding the active site (Fig. 6B). These hydrophobic regions include two loops containing the sequences PPFGF and FW as well as a helix with the sequence LPFWVRLFFW. Similarly to the loops in the B4GALNT1 AF2 model, these loops are predicted with relatively low confidence, based on pLDDT scores, suggesting they may be flexible (Supplementary Fig. S10). MD simulations of the lumenal domain demonstrate the clear association of ST3GAL5 with the membrane and insertion of these hydrophobic residues into the membrane (Fig. 6C–E).

This observation suggests that the use of non-catalytic hydrophobic surface loops may be a more general mechanism for accessing membrane-embedded lipid substrates for headgroup modification. To test this, structure predictions and MD simulations of the lumenal domains of additional ganglioside synthetic enzymes were carried out. The enzymes ST8SIA1 and B3GALT4 that process the products of

ST3GAL5 and B4GALNT1 also possess hydrophobic surface regions that can interact with PC membranes (Fig. 6A, C–E). Interestingly, the glycosyltransferases that process substrates with longer glycan chains, such as ST3GAL2 and ST8SIA5, are not predicted to interact with membranes (Fig. 6A, C–E). These data suggest that the presence of hydrophobic surface loops that insert into a membrane is positively correlated with the need to process lipid substrates with smaller headgroups.

## DISCUSSION

Here, we have determined novel experimental structures of a GSL synthetic enzyme and demonstrated that B4GALNT1 forms a homo-dimer with two dynamically remodelled active sites that change conformation upon binding UDP-GalNac substrate. These active sites are flanked by hydrophobic surface loops that contribute to the processing of lipidated substrates via insertion into the membrane bilayer. This structural arrangement of flexible hydrophobic loops surrounding the active site appears to be a conserved mechanism in other GSL synthetic enzymes.

The disulphide bonds, C80-C412 and C82-C529, near the N-terminal surface loop containing F93 have been identified previously to mediate dimer formation, consistent with the non-reducing SDS-PAGE data shown here (Fig. 1E & [19]). These were also predicted to be intermolecular disulphide bonds in the AF2 model. However, in the crystal structure, due to the alternative mainchain threading observed at residues N142-P144 (Fig. S4), these disulphide bonds become intramolecular. Although we are unable to reconcile the SDS-PAGE data with the electron density data, the symmetry of the dimer means that this has no impact upon which residues form the active site, interact with substrates or insert into the membrane. The potential importance of these disulphide bonds for enzyme function is likely to

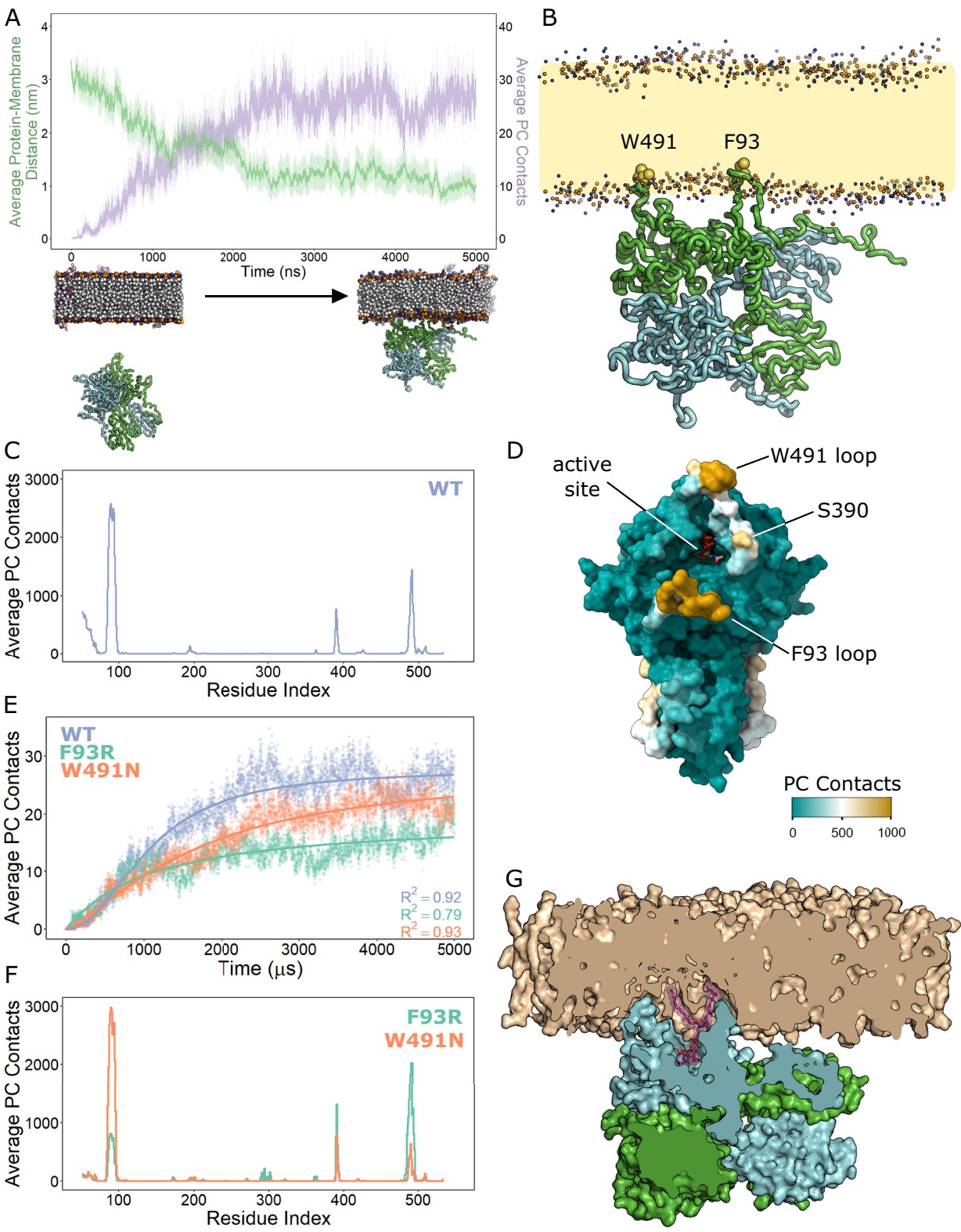

be the stabilisation of the F93 loop. MD simulations of the B4GALNT1 dimer demonstrate that when one active site is associated with the membrane, the other active site faces away from the membrane. This means that the residues between the transmembrane helix and the active site must span different distances in each molecule of the dimer. This requires one molecule to adopt a more extended conformation from the membrane to the N-terminus of the folded lumenal domain (Supplementary Fig. S11). This conformational rearrangement may result in increased tension on the N-terminus that could unfold the F93

loop if the disulphides were not present. Having demonstrated the critical importance of this loop for full activity of the enzyme against lipidated substrates, we propose that the structural constraints conferred by the disulphide bonds help maintain this loop in the appropriate conformation to associate with membranes, facilitating substrate processing.

The crystal structures demonstrated that the loop over the active site containing Y501 and R505, that binds substrate, is only ordered in the presence of UDP-GalNac. However, the continuation of this loop,

**Fig. 5 | Molecular dynamics (MD) simulations demonstrating B4GALNT1 membrane association and hydrophobic loop insertion. A** Coarse-grain MD simulations were carried out using the X-ray structure of the B4GALNT1 lumenal domain with disordered loops modelled based on AF2 predictions. For 25 independent repeats, the distance of the protein from the membrane (green) and the number of lipid (PC) contacts (purple) were quantified for each frame over the 5 µs simulation (top). The mean contacts per frame are plotted with the translucent ribbon representing the standard error of the mean. Examples of starting and final poses are illustrated (below). **B** Illustration of a final pose of the B4GALNT1 dimer interacting with the PC membrane. The orientation of the B4GALNT1 lumenal domain consistently resulted in the insertion of the F93 and W491 loops of one chain of the dimer into the PC membrane. **C** Per residue analysis of membrane lipid (PC) contacts across all simulations highlights the loops at F93 and W491 as contributing the majority of these contacts with further contribution by residues near S390. **D** Mapping of the averaged (mean) PC contacts onto the B4GALNT1 structure illustrates that the residues interacting with the membrane surround the active site. **E** The number of lipid (PC) contacts were quantified for each frame of the 5 µs MD simulations of the WT (blue), W491N (orange) and F93R (green) variants of B4GALNT1. The curves were fit to a three-parameter log-logistic model with $R^2$ values shown. **F** Per residue PC contact analysis of MD simulations using the surface loops mutants F93R and W491N. **G** Cross-section through the B4GALNT1 dimer following all-atom MD simulations illustrating GM3 (purple sticks) binding deep in the enzyme active site and the W491 and F93 loops buried in the membrane.

encompassing the hydrophobic stretch containing W491, that binds membranes, remains completely disordered in all structures. The importance of this extended, partially ordered region of the structure for B4GALNT1 function is clear from the loss of activity for mutations R505H and W491N. The structural flexibility of this region, combined with the membrane insertion of this loop, represents a dynamic mechanism allowing for binding and release of donor and lipid substrates, similar to that seen for glycoprotein-transferases[33]. When considering the roles of the two hydrophobic surface loops, it appears that the structural constraints on the F93 loop, via disulphide bonding, and the structural flexibility of the W491 loop combine to allow efficient membrane insertion while also facilitating access to the different hydrophilic GSL headgroups. The MD simulations of B4GALNT1 variants demonstrates that mutation of the F93 loop has a greater impact on membrane association (Fig. 5E). This may suggest that the F93 loop is more important for initial/sustained membrane insertion, while the W491 loop plays a role in donor substrate orientation in the active site. This is consistent with the structural data showing that the W491 loop remains disordered even in the presence of UDP-GalNac, while the F93 loop is stabilised via disulphide bonds. It is unclear whether the increased activity observed for the F93W mutation is due to enhanced affinity for the acyl chains of the ganglioside substrate or improved membrane association generally.

The MD simulations consistently demonstrate that only one active site can interact with the membrane at any one time. In the simulations, once the hydrophobic loops have inserted into the membrane, the protein does not 'roll over' to allow the other site to access the membrane. The per residue quantification of PC contacts demonstrates that the non-active site faces of the enzyme do not associate with the membrane (Fig. 5C). These non-interacting surfaces possess the three N-linked glycosylation sites, which although they are not present in the simulations, suggests they could contribute to the relative membrane orientation of B4GALNT1 by sterically hindering association of the non-catalytic surfaces.

The activity assays and MD simulations demonstrate that the lumenal domain alone can associate with membranes and process lipid substrates. This may be biologically relevant as B4GALNT1 can be cleaved at the C-terminal end of its transmembrane domain to release the lumenal ectodomain for secretion. Using N-terminal sequencing, Jaskiewicz et al.[34] showed that Golgi-resident B4GALNT1 is cleaved in/adjacent to its transmembrane helix, resulting in secretion of the B4GALNT1 ectodomain[34]. Two more recent studies using secretomics and N-terminomics of Signal Peptide Peptidase-Like 3 (SPPL3) knockout cells demonstrate that B4GALNT1 may be an SPPL3 substrate[35,36]. SPPL3 cleaves Golgi-resident type II membrane proteins, particularly glycosyltransferases, to promote their secretion, giving SPPL3 an important role in regulating the glycome of cells[35,37]. Given our work demonstrating that the B4GALNT1 lumenal domain is sufficient for activity on lipid substrates, this presents the possibility that secreted B4GALNT1 could enable the remodelling of cell surface GSLs of both the expressing cell and adjacent cells. In support of this, the UDP-substrates are also present in the extracellular space[38], and GSL composition is known to be dynamically modified by enzymes present at the plasma membrane[39,40]. Given the importance of the cell surface ganglioside profile for cell identity and ganglioside-protein interactions, remodelling of cell surface GSLs by B4GALNT1 could function as a signalling mechanism, however this is yet to be explored.

Structural and MD simulation analyses of additional ganglioside synthetic enzymes demonstrated that enzymes that synthesise the larger branched GSL headgroups, including ST8SIA5 and ST3GAL2, do not appear to possess substantial hydrophobic surface loops and are not predicted to interact with membranes. In contrast, both B4GALNT1 and ST3GAL5, which do insert into membranes, process the substrate LacCer, which possesses a small headgroup consisting of only two glycans, glucose and galactose. Analysis of the all-atom MD simulations of B4GALNT1 with GM3 substrate illustrates that the hydrophobic loops can make substantial contact with the acyl chains of the ganglioside substrate (Fig. 5G). The capacity to do this may confer greater binding affinity for substrates with smaller glycan headgroups and/or may function as a 'scoop' to help push short glycan headgroups into the active site.

To synthesise even shorter GSL headgroups, it appears that more extreme membrane association/disruption may be required. The first enzyme in the pathway, UGCG, that attaches glucose directly to ceramide, is a multi-pass, integral membrane protein. The second enzyme in the pathway, B4GALT5/6, that synthesises LacCer, is unlike the other enzymes in the pathway as it is predicted to possess a long helix, encompassing 46 residues and an extended disordered stretch of 20 residues before the catalytic domain. Future experimental structures and MD simulation studies will be required to understand how this enzyme interacts with membranes and whether this additional helix facilitates access to the small GlcCer substrate.

The observation that B4GALNT1 and other related enzymes can interact with PC membranes in the absence of their substrates may represent a mechanism for ensuring they only process lipidated substrates. By inserting hydrophobic loops/helices into the membrane, the active site remains close to the bilayer surface, restricting access to glycolipids rather than glycoproteins. In support of this, some enzymes that process longer glycan headgroups, such as ST3GAL3, with lumenal domains that do not associate with membranes, can also process glycoprotein substrates[41,42].

The structure of B4GALNT1 allows for the prediction and testing of competitive inhibitors as potential therapeutics in diseases where overexpression is driving pathogenesis (such as cancers) or where GM2 substrate reduction would be beneficial (GM2 gangliosidoses). However, it is worth noting that the membrane insertion of B4GALNT1 surface loops as part of the catalytic mechanism suggests that soluble inhibitors will need to be high-affinity to compete with the avidity advantage of membrane-embedded substrates. Future research will be required to determine whether the dynamic remodelling of the B4GALNT1 substrate binding pocket will make inhibition of this enzyme more challenging or if it will allow the binding of relatively large, potentially more specific inhibitors.

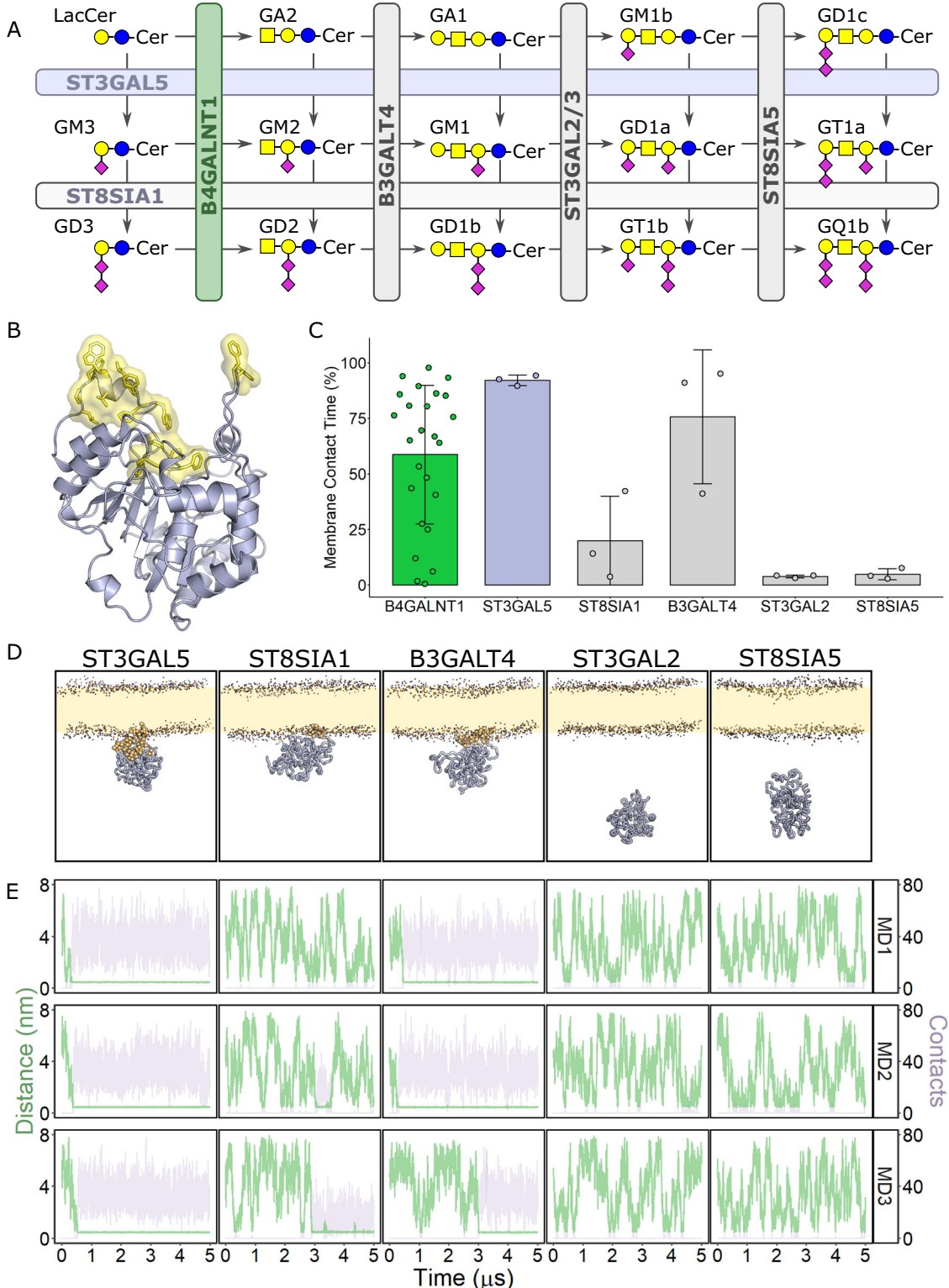

**Fig. 6 | Additional ganglioside synthetic enzymes are predicted to use hydrophobic surface loops flanking the active site to access membrane-embedded lipid substrates. A** Simplified schematic diagram of ganglioside synthesis illustrating the reactions of the enzymes analysed in panels (**B–E**), including the relevant GSL substrates and products of each reaction. **B** Ribbon diagram of the AF2 model of ST3GAL5 with hydrophobic surface loops highlighted (yellow sticks and surface). **C** Quantification of membrane contact time during 5 μs coarse-grain MD simulations for the lumenal domains of selected ganglioside synthetic enzymes. Bar height represents the mean values with error bars showing +/− SD. This quantification is from $N = 25$ for B4GALNT1 and for the $N = 3$ for the additional simulations illustrated in panel (**E**). **D** Examples of final poses following coarse-grain MD simulations for lumenal domains of ganglioside synthetic enzymes. Residues that averaged > 500 PC contacts over three 5 μs simulations are highlighted (gold spheres). **E** Plots of the distance from the membrane (green) and the number of lipid (PC) contacts (purple) for each frame of the three 5 μs simulations (MD1-3). The data for the three simulations are for the protein domains as designated in panel (**D**).

In summary, we have determined novel structures of a GSL synthesising enzyme and demonstrated that the activity of this enzyme against lipid substrates requires the insertion of hydrophobic surface loops into the membrane. This mechanism appears to be conserved in other GSL-synthesising enzymes despite very low structural similarity. Interestingly, all these enzymes are also tethered to the membrane via N-terminal transmembrane helices, highlighting that being tethered to the membrane is not sufficient for these enzymes to access and process relatively small glycan headgroups.

## Methods

### Molecular biology

The pHLSec vector for transient protein expression in HEK293-F cells was modified to encode an N-terminal $His_{10}$ tag and to remove the N-terminal secretion signal. The codon-optimised sequence for the canonical full-length human B4GALNT1 gene was designed in the pUC19 cloning vector and purchased from Invitrogen GeneArt Gene Synthesis. Primers were designed for the PCR amplification of two different N-terminal truncations of B4GALNT1 and cloned into the N-$His_{10}$-pHLSec vector using *KpnI* and *XhoI* restriction sites. For the production of stable cell lines, the ΔN46 construct of NHis10-B4GALNT1 was subcloned from the pHLSec vector into the PB-T vector via PCR amplification followed by restriction cloning with *SpeI* and *AscI* restriction enzymes. Ligation, transformation and DNA amplification was performed in *E. coli* DH5α cells. Site-directed mutagenesis was performed in the original pUC19 vector using QuikChange protocols and Pfu Turbo polymerase. The mutant B4GALNT1 constructs were then subcloned into the pHLSec vector for transient protein expression. All constructs were sequence verified. All primer sequences are provided within the Source Data.

### Protein expression and purification

HEK293-F cells (ATCC) were grown at 37 °C in 8% $CO_2$ and 130 rpm in Freestyle293 media (Gibco). For transient transfection, cells were grown to $1 \times 10^6$ cells /mL and expression constructs in pHLSec were transiently transfected using the PEI MAX transfection reagent at a 1:1.5 mass ratio of DNA:PEI, and as per the manufacturer's instructions. Wild-type and mutant proteins for activity assays were produced in this way, typically using 50 mL culture volumes and 50 µg of DNA. Proteins were expressed for 4 days, and the media was harvested by centrifugation at room temperature, beginning with $100 g$ for 5 min to gently pellet and remove the cells, followed by a $4000 g$ for 10 mins to further clarify the media before filtering through a 0.2 µm nylon membrane (Millipore). A stable expression line for the wild-type enzyme was produced by co-transfecting a 1:1:5 mass ratio of PB-RN, PBase and PB-NHis10-B4GALNT1 plasmids using MAX transfection reagent (Invitrogen) and OptiProSFM (Invitrogen) as described in ref. 43. Following selection with 500 µg/mL geneticin, protein expression was induced with 2 µg/mL doxycycline and grown and harvested as above.

Cleared media was incubated with Ni-NTA beads (Qiagen) at 4 °C on a rolling bed for 1 h. The Ni-NTA beads were captured using a gravity flow Econo-column (Biorad) and washed with wash buffer containing 20 mM Tris pH 7.4, 150 mM NaCl, 20 mM imidazole and 1 mM $MnCl_2$. Protein was eluted in elution buffer containing 20 mM Tris pH 7.4, 150 mM NaCl, 300 mM imidazole and 1 mM $MnCl_2$. The eluted protein was further purified via size exclusion chromatography in buffer containing no imidazole using a Superdex 200 Increase 10/300 column (Cytiva).

### Sodium Dodecyl Sulphate Polyacrylamide Gel Electrophoresis (SDS-PAGE)

Protein samples were denatured in either reducing (with 50 mM Dithiothreitol) or non-reducing SDS loading dye (40 mM Tris pH 6.8, 2% SDS, 4% glycerol, 0.01% bromophenol blue) followed by heating for 5 min at 95 °C. Pre-poured NuPAGE 4–12% Bis-Tris polyacrylamide gradient gels were run in 1X MOPS SDS running buffer (NuPAGE) for 50 min at 200 V. Gels were run in a Bio-Rad Mini-PROTEAN electrophoresis cell at room temperature. Post electrophoresis, gels were incubated with Coomassie InstantBlue (Abcam) on a rocking bed prior to washing and de-staining in water. Gels were imaged using a BioRad Chemidoc™ MP Imaging system set to auto-optimal exposure.

### Crystallisation

Purified $His_{10}$-ΔN46-B4GALNT1 was concentrated using a 30 kDa MWCO centrifugal concentrator (Millipore) to 6 mg/mL. Crystallisation experiments were conducted in 96-well nanolitre scale sitting drop vapour diffusion plates (200 nL protein:200 nL reservoir) using a Mosquito LCP Crystallisation Robot. The plate was incubated at 20 °C against 80 µL of reservoir solution. Original crystals were grown against a reservoir of 30% GOL_P4K (40% v/v glycerol, 20% w/v PEG 4000), 0.1 M Tris-BICINE pH 8.5, 0.2 M sodium formate, 0.2 M ammonium acetate, 0.2 M sodium citrate tribasic dehydrate, 0.2 M potassium sodium tartrate tetrahydrate and 0.2 M sodium oxalate. Diffraction-quality crystals were grown following microseeding as described previously[44] using a seed stock prepared from original crystals and combined with purified protein at 6 mg/mL. For the Apo structure, crystals underwent cryoprotection in a reservoir solution supplemented with 20% (v/v) glycerol and 5 mM $MnCl_2$ before flash-freezing under liquid nitrogen. For the UDP-GalNac bound structure, crystals underwent cryoprotection in a reservoir solution supplemented with 20% (v/v) glycerol, 5 mM $MnCl_2$ and 5 mM UDP-GalNac (Promega) and were soaked for 5 min before flash-freezing under liquid nitrogen. For the UDP-bound structure, crystals underwent cryoprotection in a reservoir solution supplemented with 20% (v/v) glycerol, 5 mM $MnCl_2$ and 2 mM UDP (Promega) and were soaked for 4 min before flash-freezing under liquid nitrogen.

### X-ray Data Collection and Structure Determination

Diffraction data were collected at Diamond Light Source (DLS) at beamline I04 for the Apo structure and I24 for the UDP-GalNac and UDP-bound structures on a CdTe Eiger2 9 M detector at 100 K (Table 1). Data were indexed and integrated using DIALS[45] via the Xia2 automated data processing pipeline at DLS and were scaled and merged using AIMLESS via CCP4 Cloud[46,47]. Structure solution was performed with molecular replacement using Phaser with an AF2-Multimer prediction of the ΔN46-B4GALNT1 homodimer as the search model. This model was generated using default parameters in ColabFold[48]. The structure was iteratively refined using Refmac, COOT[49] and ISOLDE[50] implemented in ChimeraX 1.6[51]. MolProbity statistics were used for validation during refinement. Disordered loops were removed and glycans added manually in COOT. Ligands were built using the CCP4 ligand dictionary in COOT. Graphical figures were rendered in PyMOL (Schrödinger, LLC) and ChimeraX.

### Differential scanning fluorimetry

Differential scanning fluorimetry was performed using the Protein Thermal Shift Dye kit (Thermo Fisher Scientific) as per the manufacturer's instructions using a Bio-Rad CFX96™ real-time PCR detection system. Reaction mixes consisting of 6 µg recombinant protein and 5X SYPRO orange dye were made up to a total volume of 25 µl in purification buffer and transferred to a white V-shaped 96-well qPCR plate (Bio-Rad). Samples were heated on a 0.5 °C/s gradient from 10 to 95 °C, and protein unfolding at each temperature monitored by measurement of fluorescence at 580/623 nm (excitation/emission). Normalised fluorescence against temperature was fitted to a non-linear Boltzmann sigmoidal distribution in GraphPad Prism 5, and melting temperatures calculated from the inflection point.

### Liposome preparation

Lipids were purchased from Merck Avanti as follows: phosphatidylcholine PC (#840051 C), GM3 (#860058 P), GD3 (#860060 P),

LacCer (#860598 P), DGS-NTA(Ni) (#790404P). Liposomes were prepared by first dissolving powdered lipid stocks in either 1:1 (GD3) or 2:1 (GM3/LacCer) chloroform:methanol to give 2.5 mg/mL stock solutions of ganglioside or 10 mg/mL solutions of PC. Liquid ganglioside and PC stocks were combined in glass vials and mixed using a glass syringe. Lipid solutions were then dried down in a Savant Speedvac SPD140 at 9 Torr and 45 °C. The resultant lipid film was then hydrated using activity assay buffer (25 mM Tris pH 7.4, 5 mM $MnCl_2$, 2.5 mM $CaCl_2$ and 12.5 mM NaCl) to form multilamellar liposomes. The final liposome concentration was 10 mM, with variable ganglioside, Ni-NTA(DGS) and PC concentrations depending on the experiment.

## Activity assays

All B4GALNT1 activity assays were conducted either in low-bind Eppendorf tubes or in 96-well clear, flat-bottomed plates. For all assays, 60 μl of B4GALNT1 solution was combined with 40 μl of substrate solution and incubated either on an end-over-end rotator (for tubes) or a plate shaker (for plates) at room temperature for 15 min. Depending on the assay, the concentration of B4GALNT1 used ranged from 0 to 500 nM, and the composition and concentration of substrates is as described in the main text. Assays were measured by transferring 25 μl of the reaction mixture into a V-bottomed white 96-well qPCR plate (Bio-Rad) for the UDP-Glo™ (Promega) step, in which 25 μL of UDP detection reagent is added to each condition, quenching the glycosyltransferase reaction. The plate was then mixed on a plate shaker for 2 min at room temperature before incubating on the bench for 20 min. Luminescence was subsequently measured with a Clariostar (BMG LabTech) plate reader as an endpoint. Ideal assay conditions for endpoint analysis were identified through enzyme-titration coupled time-course experiments (Supplementary Fig. S2B).

## Coarse-grained molecular dynamics (MD) simulations

For monitoring the membrane-association of the lumenal domain of B4GALNT1, missing loops from the crystal structure were rebuilt based on an AF2-predicted model aligned to the crystal structure in COOT. For simulations with loop mutations, the amino acid changes F93R and W491N were made to these models using COOT. Coarse-grained MD simulations were conducted with GROMACS 2023.4[52]. B4GALNT1 membrane association simulations were prepared using the MemProtMD-Association component of the MemProtMD pipeline[53], generating 25 repeats with different protein starting orientations. 5 μs production simulations were performed with a preformed 1% GM3, 99% PC membrane assembled with INSANE 3.0[54]. Coarse-grained models were generated using the Martini v3.0.0 force field[55].

For coarse-grained simulations of membrane-embedded B4GALNT1, the N-terminal domain, transmembrane helices and linker not present in the crystal structure (residues 1–50) were grafted onto the model based on an AF2 prediction[48,56] aligned in COOT to the model from the association study[49].

Coarse-grained membrane-embedded simulations of B4GALNT1 were prepared using the MemProtMD-Insane component of the MemProtMD pipeline[53]. Memembed[57] was used to position the transmembrane helices in the membrane, and a preformed symmetrical 1% GM3, 99% PC membrane was assembled with INSANE 3.0[54].

For membrane-association simulations of the other lipid synthases, AF2 models were truncated from the N-terminus to remove transmembrane helices and residues with low pLDDT scores upstream of the folded domain. The starting residues for these models were: D101 for ST3GAL5; R50 for ST8SIA1; A52 for B3GALT4; K68 for ST3GAL2, and F53 for ST8SIA5. For ST8SIA1, nine residues were also truncated from the C-terminus based on poor structural predictions (low pLDDT score). Selected residues were removed in ChimeraX 1.6[51]. These lipid synthase membrane association simulations were prepared using the MemProtMD-Association component of the MemProtMD pipeline[53], generating 3 repeats. Simulations were run for 5 μs with a preformed 100% PC membrane assembled with INSANE 3.0[54]. Coarse-grained models were generated using the Martini v3.0.0 force field[55].

All coarse-grained systems were energy minimised via steepest descent. Systems were equilibrated for 20 ns through 0.01 ps time steps with semi-isotropic Berendsen pressure coupling and V-rescale group temperature coupling (Berendsen barostat and V-rescale thermostat). 5 μs/2 μs production runs were performed through 0.02 ps time steps using V-rescale temperature coupling and C-rescale semi-isotropic pressure coupling. All coarse-grained simulations utilised the LINCS algorithm for bond length constraints.

## All Atom Molecular Dynamics

For all-atom simulations of B4GALNT1, representative GM3-bound poses from the coarse-grain simulations was identified using the PyLipID software[29]. Two binding sites were selected based on residence time, occupancy, surface and duration statistics. The two sites were equivalent, representing the same site for chain A vs chain B. The top-ranked (most representative) pose for this binding site was chosen, and the corresponding frame was converted from a coarse-grain representation to atomistic using CG2AT[30] with alignment to the original model (the crystal structure with modelled loops). CG2AT performs an initial NVT (number of particles, volume and temperature) equilibration. A 10 ns equilibration simulation was performed before a 500 ns production run, both through 0.002 ps time steps with V-rescale temperature coupling and C-rescale semi-isotropic pressure coupling. Bond length constraints were implemented through the LINCS algorithm. All simulations were performed using GROMACS 2023.4.

## Data analysis and visualisation

To quantify protein lipid-interactions and protein-membrane distance, either GROMACS tools (gmx mindist) or PyLipID analysis were used. When using gmx mindist, a 0.6 nm distance cut-off was used for CG simulations. To calculate lipid contacts per residue for the CG simulations, an in-house Python script was used to convert gmx mindist outputs. In all cases, the plotted MD data represents the averaged (mean) result for all repeats. For PyLipID analysis, dual distance cutoffs of [0.5, 0.8] and [0.4, 0.6] were used for CG and atomistic simulations, respectively. All activity assay and molecular dynamics data were plotted and analysed in R with structural figures prepared in either PyMOL 2.6 (Schrödinger, LLC) or ChimeraX 1.6.1[51].

## Reporting summary

Further information on research design is available in the Nature Portfolio Reporting Summary linked to this article.

# Data availability

The atomic coordinates and structure factors have been deposited in the Protein Data Bank (PDB) under accession codes 9H6J (Human B4GALNT1 Apo Structure), 9H6K (Human B4GALNT1 in Complex with UDP-GalNac), 9H6L (Human B4GALNT1 in Complex with UDP). Experimental data for activity assays, size exclusion and thermal shift experiments, primers, as well as PDB validation reports and MD simulation analysis are available in the Source Data file. MD simulation data files and trajectories have been deposited in the University of Cambridge Data Repository [https://doi.org/10.17863/CAM.116886]. Source data are provided in this paper.

# Code availability

MD simulation data files and trajectories have been deposited in the University of Cambridge Data Repository [https://doi.org/10.17863/CAM.116886].

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

## Acknowledgements

We acknowledge Diamond Light Source for time on beamlines I04 and I24 under proposals MX28677 and MX36838. We thank CIMR IT and, in particular, Nic Mitchell, for help getting MD and analysis software running locally. This work was supported by a Wellcome Trust Senior Research Fellowship (219447/Z/19/Z) to JED. JWJW is supported by a Michael House PhD studentship. PJS's lab was funded by Wellcome (208361/Z/17/Z), MRC (MR/Z504245/1), BBSRC (BB/Y007603/1; BB/Y003187/1; BB/Y003306/1), EPSRC (EP/Z535709/1), NIH (R01AI176776; R01AI174416), JPIAMR and the Howard Dalton Centre. This project made use of time on ARCHER2 granted via the UK High-End Computing Consortium for Biomolecular Simulation, HECBioSim (http://www.hecbiosim.ac.uk), supported by EPSRC (grant no. EP/R029407/1). PJS would like to thank the SCRTP at Warwick for the use of the computing infrastructure.

## Author contributions

J.E.D. and J.W.J.W. conceived the study. J.E.D., J.W.J.W., H.G.B. and P.J.S. designed experiments. J.E.D. and J.W.J.W. performed structural biology experiments and refined structural models. J.W.J.W. performed biochemistry experiments and molecular dynamics experiments. J.E.D. and J.W.J.W. analysed the results. J.E.D. and J.W.J.W. wrote the manuscript.

## Competing interests

The authors declare no competing interests.
