## [Transparent Peer Review file · Nature Communications]

Conformational Dynamics and Membrane Insertion Mechanism of B4GALNT1 in Ganglioside Synthesis

Corresponding Author: Professor Janet Deane

Version 0:

Reviewer comments:

Reviewer #1

(Remarks to the Author)

This manuscript from the Deane lab reports the first structure of human B4GALNT, an important enzyme in synthesizing gangliosides with relevance to neurodegenerative disease as mutations in B4GALNT result in HSP26. The authors generate soluble forms of B4GALNT by removing the single pass transmembrane helix and characterize enzymatic activity using both a soluble substrate and a membrane bound lipid substrate. Activity is weak but becomes robust in a liposome system when B4GALNT is recruited/tethered to liposomes using Ni-NTA lipids via the recombinant His-tagged B4GALNT. Intermolecular disulfide bonds appear present in solution, but the homodimeric crystal structure suggests intramolecular disulfide bonds. Structures are determined for apo enzyme, as well as substrate and product bound. Comparing these structures reveals ordering and rearrangement of loops around the active site. Consistent with the hypotheses generated from the structures, point mutations of residues that contact substrate or product reduce activity. The structure also allows analysis of disease-associated missense mutations in B4GALNT. The authors classify variants and identify two unclassified variants to result in a major reduction in activity. The authors further use a combination of MD simulations and biochemical analysis to identify two loops with hydrophobic residues that mediate membrane anchoring and are important for B4GALNT activity. Lastly, they expand this analysis to other enzymes in the glycosphingolipid biosynthetic pathway.

Overall, the manuscript is well-written and makes noteworthy advances that will be appreciated by those interested in neurodegeneration, lipid metabolic enzymes, and structural biology. The statistics for the crystals structures indicate they are well-refined. There are just a few minor points below that the authors could try to address to improve the manuscript.

1. Supplementary Figure 2. The Y-axis labels only say pmol/hr and the figure legend is not detailed. The authors should specify what the pmols in the Y-axis are. I assume it is pmol of substrate hydrolyzed or pmol of product produced.
2. The authors say the activity assays did not follow standard Michaelis Menten kinetics. They make the conclusion that B4GALNT prefers GM3 or GD3 as substrates based on higher activity. This conclusion seems reasonable. Although the system using Ni-NTA lipids may preclude this, have they tried to verify their conclusion by analyzing activity as a function of the surface concentration of GM3 and GD3 in liposomes? Here they would keep the bulk concentration of GM3/GD3 constant (e.g. 100 μ M) and vary the concentration of phospholipids used to create liposomes. This would vary the mol% of GD3 or GM3, and the authors can check if B4GALNT may prefer one of the lipid substrates as a function of lipid substrate surface concentration.
3. Disulfide threading. This part remains confusing and unclear. Authors provide evidence that recombinant protein forms intermolecular disulfide bonds in solution (Fig. 1E), as they state, the crystal structure shows clear electron density for only intramolecular disulfide bonds. While the authors have clearly thought about this deeply, and there is some discussion made of this point, is there any additional evidence or thoughts that they have to explain this discrepancy?
4. The authors could consider mentioning it is human B4GALNT in the abstract and the initial mention in the results for clarity.
5. In Fig. 1D, the residue number for the luminal domain ends at 355. That does not seem correct. Can they clarify or update?

6. Page 8. In the sentence “the three variants predicted to be disease-causing ...” Is D358A a disease-associated variant? It was previously mentioned it was a catalytically dead variant but not disease associated.

Mike Airola

Reviewer #2

(Remarks to the Author)

The work contributed by Welland and colleagues represents a significant advance in our understanding of glycosphingolipids biosynthesis, also providing remarkable mechanistic insights into the integral membrane B4GALNT1 and related homologues. Specifically, the authors report three crystal structures of the human dimeric truncated version of B4GALNT1, His10- Δ N46-B4GALNT1, in (i) its unliganded form, and in complex with (ii) UDP-GalNac and (iii) UDP, at 2.94Å, 2.75Å and 2.67Å resolution, respectively. Based on structural, biophysical and enzymatic/mutagenesis experimental data, further extended/interpreted by molecular dynamics simulations, a molecular mechanism for substrate binding/product release and catalysis is proposed. The manuscript is well written and advances the field. Because of its novelty it is therefore suitable for publication in Nature Communications.

Please, find below some comments/suggestions for improvement:

1. “an N-terminal region encompassing residues 130 to 257 that forms the non-catalytic domain (the “lower” domain in Fig. 2A) while the C-terminal region, residues 258-533 forms much of the predicted catalytic domain”. A good opportunity to introduce new panels in Figure 2, showing the two domains in different colors and labeling/numbering the secondary structure elements. In addition, labeling/numbering the subunits/protomers and secondary structure elements in current panels 2A and 2B would be helpful for readers.
2. “B4GALNT1 is the only validated human protein in CAZY glycosyltransferase family 12 and is an inverting GT-A type glycosyltransferase”. I suggest introducing a structure-based sequence alignment to visualize the structure/sequence conservation of amino acids involved in substrate binding/specificity and catalysis. The alignment could be further extended to other selected B4GALNT1 structural homologues, restricted to the catalytic domain.
3. “Several missense mutations of B4GALNT1 have been linked to the development of HSP26 (Supp. Table 1).” I suggest moving this Table to the main text.
4. “Analysis of the NCBI ClinVar [26] and gnomAD [27] databases, identifies over 100 different missense variants that have been identified in patients with spastic paraplegia but are currently of uncertain significance.” I suggest elaborating a Supplementary Table, showing the location of each mutant in the context of the Δ N46-B4GALNT1 structure. Is there any of those mutations located into the transmembrane alpha-helix?
5. “What remains unclear is how the acceptor lipid substrate, GM3 or GD3, binds the enzyme.” Based on the experimental structure of the Δ N46-B4GALNT1 dimer, could the transmembrane alpha helices also dimerize?

Congratulations to the authors for this very nice work.

Reviewer #3

(Remarks to the Author)

The work presents diffraction implied structures (Figs. 2 and 3) of the B4GALNT1 dimer that is a major enzyme in ganglioside synthesis. Structures for reactant and product indicate that the active site changes conformation (Figure 2). SDS-PAGE indicates the dimer fraction population is unstable when disulfide bonds are broken (Fig. 1E). Activity of membrane localized (via his-tag/Ni-NTA) and soluble forms of the catalytic domain are interesting and logically consistent with the mutually membrane co-localized forms of enzyme and substrate highly active (Fig 1F).

Molecular simulations (my expertise) employing coarse grained models show that dynamic loops with functionally important (experiment, Fig 4) hydrophobic residues embed in the membrane, with only one molecular chain of the dimer interacting. Captured coarse-grained GM3 lipids are converted to all-atom which are stable over a short period of time (500 nanoseconds, supplemental movie). The simulations are a good addition to the paper as they illustrate the broad molecular orientations and interactions of the wild type and mutant. It is a minor point that does not truly affect the message of the paper, but the simulations do not discriminate the energetic interactions present as they only provide the time for developing membrane interactions given a non-equilibrium starting condition (e.g., Figure 5A). Figure 6D illustrates the weakness of the approach: ST3GAL2 and ST8SIA5 are judged not to bind yet they only attempt a few interactions with the membrane (compare ST8SIA1 which has a few unsuccessful interactions before finally binding). Statistical analysis may be possible for these systems that would transform the data from qualitative to quantitative but is not strictly necessary for publication in my view --- I would like to see some discussion of why statistical analysis could not be performed, given the simulation approach. For example, were the insertion depths of the mutant and wild type not significant?

The paper leaves inconclusive how the SDS-PAGE experiments indicate intermolecular disulfide bonds yet the crystal structure indicates intramolecular. The varied threading of the AF2 and crystal models may represent nearly degenerate

alternative folds that are both present.. the electron density does not unambiguously rule out a population of the AF2-threaded structure. It is hard to see from the single view in Figure S3 and without the pdb file, but the assignment of N142 appears difficult, considering the N142-A contact to the chain B backbone carbonyl appears to repulsive. It is not my expertise, but if the SDS conditions allowed intramolecular disulfide bonds to reform as intermolecular in the conditions before electrophoresis it would be hard to discern them from this experiment. In the absence of further targeted biochemistry I am inclined to trust the crystal structure over the messy SDS conditions. I did not see SDS-PAGE experiments mentioned in the methods.

The paper is cleanly written and well edited to my standards.

In summary, I enjoyed reading this paper, which I believe has the right mix of thought provoking links between protein structure and detailed function. The inconclusive threading of the two molecular chains is more interesting than problematic. The identification of important hydrophobic residues and their identification in similar enzymes is important, especially considering the context established by the possibility of the enzymatic domain being functional as a released, soluble element. The simulations provided a valuable qualitative picture of the possible orientations and importance of the hydrophobic residues. The lack of quantitative support (that would be amenable to statistical analysis) by the simulations was a weakness. As written it is likely that broader readership will overestimate the significance of the simulation findings, for example, that the all-atom form of GM3 stays in the binding pocket for 500 nanoseconds was virtually assured if the system was even modestly stable. Caveats on the significance of the simulations, and how future work (employing meta dynamics, umbrella sampling or non-equilibrium pulling) might establish quantitative differences would improve the narrative.

Minor:

Note that the all-atom simulations were 500 nanoseconds long in text. I was reassured that the all-atom simulations were possible, but this duration (500 nanoseconds) does not provide any rigorous stability of the bound GM3 in the pocket).

Mishra and Johnson have a quantitative analysis of the 2D/3D dimensional reduction that leads to greater activity for the co-localized system (<https://doi.org/10.1063/5.0045867>). This might be helpful for interpreting how the membrane-localized vs. soluble activity depends on the amount of membrane and volume of solution.

Version 1:

Reviewer comments:

Reviewer #1

(Remarks to the Author)

All of my concerns have been adequately addressed in the revised manuscript.

Reviewer #2

(Remarks to the Author)

The authors answered all my comments and kindly considered my suggestions on the manuscript.

Reviewer #3

(Remarks to the Author)

The revision addressed all of my (minor) comments from the initial submission.
